# Modulation of Membrane Trafficking of AQP5 in the Lens in Response to Changes in Zonular Tension Is Mediated by the Mechanosensitive Channel TRPV1

**DOI:** 10.3390/ijms24109080

**Published:** 2023-05-22

**Authors:** Rosica S. Petrova, Nikhil Nair, Nandini Bavana, Yadi Chen, Kevin L. Schey, Paul J. Donaldson

**Affiliations:** 1Department of Physiology, School of Medical Sciences, New Zealand National Eye Center, Faculty of Medical and Health Sciences, The University of Auckland, Auckland 1023, New Zealand; 2Department of Biochemistry, Mass Spectrometry Research Center, Vanderbilt University, Nashville, TN 37232, USA

**Keywords:** lens, AQP5, lens surface pressure, TRPV1, water transport

## Abstract

In mice, the contraction of the ciliary muscle via the administration of pilocarpine reduces the zonular tension applied to the lens and activates the TRPV1-mediated arm of a dual feedback system that regulates the lens’ hydrostatic pressure gradient. In the rat lens, this pilocarpine-induced reduction in zonular tension also causes the water channel AQP5 to be removed from the membranes of fiber cells located in the anterior influx and equatorial efflux zones. Here, we determined whether this pilocarpine-induced membrane trafficking of AQP5 is also regulated by the activation of TRPV1. Using microelectrode-based methods to measure surface pressure, we found that pilocarpine also increased pressure in the rat lenses via the activation of TRPV1, while pilocarpine-induced removal of AQP5 from the membrane observed using immunolabelling was abolished by pre-incubation of the lenses with a TRPV1 inhibitor. In contrast, mimicking the actions of pilocarpine by blocking TRPV4 and then activating TRPV1 resulted in sustained increase in pressure and the removal of AQP5 from the anterior influx and equatorial efflux zones. These results show that the removal of AQP5 in response to a decrease in zonular tension is mediated by TRPV1 and suggest that regional changes to P_H2O_ contribute to lens hydrostatic pressure gradient regulation.

## 1. Introduction

In the absence of a blood supply, the mammalianlens utilises a unique microcirculation system to maintain its overall cellular structure, which establishes its transparent and refractive properties [1,2]. Structurally, the lens consists of two cell types: an epithelial cell layer that covers the anterior surface of the lens and fiber cells that comprise the bulk of the lens (Figure 1). At the equator of the lens, these epithelial cells continuously divide to form differentiating fiber cells that exhibit extensive changes in their protein expression profile, undergo massive elongation, and eventually lose their light-scattering cellular organelles [3,4]. During the elongation process, the original membrane domains of progenitor epithelial cells migrate along the epithelium and capsule (Figure 1B), and their lateral membranes adopt a hexagonal profile that consists of two broadsides and four narrow sides (Figure 1C), which facilitates the packing of fiber cells into an orderly array [5]. This process continues until the apical and basal tips of fiber cells from the opposing lens hemisphere meet to form the anterior and posterior sutures [6], respectively. Since these processes of epithelial cell division and subsequent fiber cell differentiation continue throughout life, newly differentiated secondary fiber cells constantly internalise older mature fiber cells, which in turn have internalised the oldest primary fiber cells that were originally laid down during embryonic development [7,8].

To maintain this structure, the lens generates an internal microcirculation that utilises circulating ionic and fluid fluxes to not only control the lens water content in order to preserve fiber cell volume [9], with the water-to-protein ratio setting the lens gradient of the refractive index [10], but also to rapidly deliver nutrients and antioxidants to deeper fiber cells [11]. These circulating ionic and fluid fluxes preferentially enter the lens at the anterior and posterior poles via an extracellular influx pathway associated with the anterior and posterior sutures (Figure 1A, influx). Ions and water then cross the membranes of deeper mature fiber cells via membrane channels before moving towards the lens surface via an intracellular outflow pathway (Figure 1A, outflow) mediated by gap junction channels [12]. The subcellular alignment of gap junctions on the broadside domains of the hexagonal-shaped lateral fiber cell membranes [13] serves to direct the intercellular flow of ions and water to the lens equator [14], where Na^+^/K^+^ATPase that drives the circulating ionic current is concentrated [15]. Once returned to the surface cells of the lens, ions and water then use a variety of channels and transporters to move across the membranes of equatorial epithelial and peripheral fiber cells to leave the lens (Figure 1A, efflux zone). 

**Figure 1 ijms-24-09080-f001:**
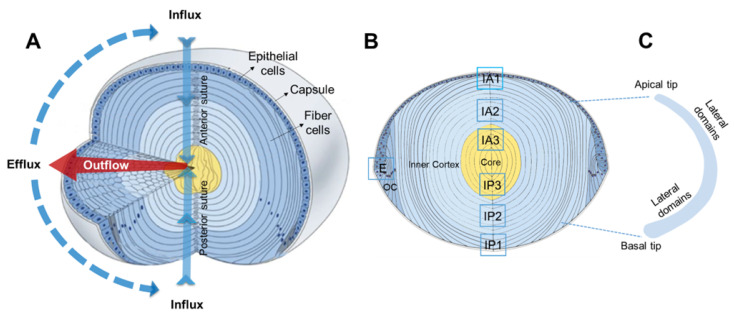
Lens anatomy and regional differences in water flow in the lens. (**A**) The biconvex lens is covered with an elastic capsule and is composed of predominantly elongated fiber cells, while its anterior region is layered with an epithelium. The regional three-dimensional representation of a microcirculation model shows ion and fluid fluxes that enter the lens at both poles via the extracellular space associated with the anterior and posterior sutures (solid light blue arrows) before crossing fiber cell membranes and exiting via an intercellular outflow pathway mediated by gap junctions (red arrow), which directs the fluxes to the equatorial efflux zone where they exit the lens. Reprinted/adapted with permission from Shi et al. [16]. Copyright year: 21 Mar 2023, copyright owner’s name: Rosica S Petrova. (**B**) Diagram of an axial section of the lens showing the relative locations where confocal images have been taken to assess the effects of modulating zonular tension and TRPV1 activation on the membrane location of AQP5 in the anterior (IA1, IA2, and IA3) and posterior (IP1, IP2, and IP3) influx pathways associated with the sutures and the equatorial efflux (E) zone. (**C**) Schematic of an isolated fiber cell depicting the apical and basal tips that form the anterior and posterior sutures, respectively, and the greatly elongated lateral membranes. OC—outer cortex.

This outward flow of water through the gap junction channels has been shown to generate a substantial hydrostatic pressure gradient [17], which is conserved in all species of lenses studied to date [18]. In the mouse lens, this pressure gradient has been shown to be regulated by a complex dual feedback system that, by reciprocally modulating Na^+^/K^+^-ATPase [19] and NKCC1 [20] activities, alters the osmotic gradients that drive water transport through the lens. This feedback system utilises the mechanosensitive ion channels, TRPV1 and TRPV4, to sense decreases and increases, respectively, in the lens pressure to activate competing arms of the dual feedback loop system to maintain a constant lens pressure. Activation of TRPV1 by capsaicin or hyperosmotic challenge has been shown to cause a biphasic increase in lens pressure, which can be sustained by blocking the TRPV4-mediated arm of the dual feedback system. Conversely, activation of TRPV4 by GSK or hyposmotic challenge causes a biphasic decrease in the lens pressure, which can be sustained by blocking the TRPV1-mediated arm of the dual feedback system [19,21]. We have also shown that TRPV1/4 is also capable of sensing changes to the tension applied to the lens by either cutting the zonules that attach the lens to the ciliary muscle or through pharmacological modulation of the ciliary muscle via the muscarinic agonist pilocarpine and antagonist tropicamide, which increases or decreases contractility, respectively [22]. Interestingly, while the effects of pilocarpine and tropicamide on zonular tension are also mediated by TRPV1 and TRPV4, respectively, they induce sustained changes to the magnitude of the hydrostatic pressure gradient and not the biphasic change observed via the direct pharmacological activation of either TRPV1 or TRPV4 [22].

Thus, it would appear that overall water transport in the lens is regulated by TRPV1/4-mediated modulation of ion transporter activity to affect local changes to osmotic gradients, which drive the passive diffusion of water into and out of the lens, across cell membranes, and between cells through gap junction channels. However, the rate of water movement via these pathways is not solely determined by the magnitude of the osmotic gradient, but also by the permeability of cell membranes or gap junction channels to water. In the case of membrane water permeability (P_H2O_), it is a product of the lipid composition of cell membranes and the complement of aquaporin (AQP) water channels expressed in plasma membranes [23]. In the lens, at least three AQPs (AQP0, 1, and 5), each with distinctly different P_H2O_ regulation and post-translational modification patterns, are differentially expressed in different regions of the lens [24,25]. AQP1, a constitutively active water channel that is widely expressed in a variety of different tissues, is only found in the epithelial cell layer. In contrast, AQP0, which expression is relatively restricted to the lens, is only found in fiber cells. Although highly abundant, AQP0 is a poor water channel that undergoes extensive post-translational modification and has been shown to also act as an adhesion protein [26]. The third major water channel in the lens is AQP5, which has been shown to be expressed in both epithelial and fiber cells [27,28].

The trafficking of AQP5 to and from plasma membranes has been shown to alter the P_H2O_ in a variety of epithelial tissues [29], including lens fiber cells [30]. More recently, the regulation of subcellular localisation of AQP5 in the rat lens has been shown to be modulated by changes to the tension applied to the lens via the zonules [31]. Reducing the tension applied to the lens by either cutting the zonules or by pilocarpine-induced contraction of the ciliary muscle resulted in the removal of AQP5 from the plasma membranes in specific regions of the lens. AQP5 labelling in the apical tips of inner cortical fiber cells that form the sutures in the anterior influx pathway and in the lateral membranes of peripheral fiber cells in the efflux zone was lost from the membranes following pilocarpine treatment [31]. Interestingly, no AQP5 labelling was found in the basal tips of fiber cells that form the sutures in the posterior influx zone, and this association was not affected by changes in zonular tension, a finding that suggests P_H2O_ and, hence, water transport are differentially regulated in the anterior and posterior influx zones [31].

Based on the abilities of pilocarpine-induced decrease in zonular tension to cause sustained increase in hydrostatic pressure in the mouse lens [22] and induce the removal of AQP5 from fiber cell membranes in the rat lens [31], we hypothesise that both processes are linked via the activation of the TRPV1 channel. In the current study, we tested this hypothesis by adopting the rat lens as our experimental model since it has proven to be more amenable than the mouse lens for the study of the trafficking of AQP5 to and from cellular membranes. This is due to the observation that, relative to the mouse lens, the rat lens has a larger zone in the outer cortex where the trafficking of AQP5 to and from cellular membranes can be more easily studied [30]. Hence, in this study, we first confirmed that, like the mouse lens, pilocarpine-induced reduction in zonular tension in the rat lens is also mediated by TRPV1, before showing that pilocarpine-induced removal of AQP5 from cellular membranes is abolished by the inhibition of TRPV1, and, in the absence of pilocarpine, it can be mimicked by the inhibition of TRPV4 and subsequent activation of TRPV1. These results suggest that modulation of the movement of water via a change to the membrane localisation of AQP5, which alters P_H2O_, may be associated with the regulation of hydrostatic pressure in the rat lens.

## 2. Results

The goal of this study was to determine whether the removal of AQP5 from the membranes of fiber cells in the anterior influx and equatorial efflux zones of the rat lens induced by pharmacological reduction in zonular tension caused by the addition of pilocarpine [31] is mediated by activation of the mechanosensitive channel, TRPV1. In the mouse lens, we had previously shown that pilocarpine addition resulted in sustained increase in lens hydrostatic pressure that was mediated via TRPV1 activation [22]. Hence, to achieve our goal, we first needed to establish whether the addition of pilocarpine to increase the contraction of the ciliary muscle and reduce the tension applied to the lens via the zonules also alters hydrostatic pressure in the rat lens before determining whether the membrane trafficking of AQP5 can be regulated by modulating the activity of TRPV1 in the rat lens.

### 2.1. Pilocarpine Increases Hydrostatic Pressure in the Rat Lens

Like we had previously observed in the mouse lens [22], we found that the application of pilocarpine to the rat lenses also produced a relatively rapid increase in hydrostatic pressure at the lens surface, which peaked around ~80 min at 14.80 ± 0.74 mmHg without returning to the baseline level throughout the duration of the experiment (Figure 2A). To further confirm that, like the mouse lens, activation of TRPV1 is required for pilocarpine-induced sustained increase in surface pressure, the rat lenses were pre-incubated with the TRPV1 inhibitor, A-88, for 20 min before pilocarpine was added to the bath in the presence of A-88 (Figure 2B). In the presence of the TRPV1 inhibitor, pilocarpine-induced increase in lens surface pressure was effectively inhibited, with the surface hydrostatic pressure only increasing by 0.35 ± 2.82 mmHg at the end of the recording period. Hence, it appears that an increase in lens surface pressure in response to a reduction in the tension applied to the lens caused by pharmacological modulation of ciliary muscle contractility is detected in both the mouse [22] and rat lenses (present study) via TRPV1 channels. Next, we determined whether the inhibition of TRPV1 affects the ability of pilocarpine to induce removal of AQP5 from the membranes of fiber cells that form the anterior and posterior sutures.

### 2.2. Effect of Inhibition of TRPV1 on Membrane Trafficking AQP5 in the Anterior and Posterior Influx Zones

In our previous study [31], we used axial sections to map the subcellular distribution of AQP5 in the equator efflux zone and in the anterior and posterior influx pathways, in three representative zones, in the presence and absence of pilocarpine (Figure 3A). As seen on Figure 3, AQP5 does not colocalise to the posterior suture in the outer cortex (IP1) or the inner cortex (IP2) in either the absence (Figure 3B, IP1 and IP2) or presence (Figure 3C, IP1, IP2) of pilocarpine, although it is associated with the sutures in the core (IP3) of the lens (Figure 3B,C, IP3). We also showed that preincubation of the lenses in the TRPV1 inhibitor A-88, followed by the application of pilocarpine, does not change the localisation of AQP5 at the basal tips of the posterior suture (Figure 3D, IP1, IP2, and IP3). In contrast, pilocarpine has a specific effect on the membrane localisation of AQP5 in the inner cortical area (IA2) of the anterior suture (Figure 4). Under the control conditions in which the in situ zonular tension is preserved, AQP5 is absent from the apical tips of fiber cells that form the suture in the outer cortex (Figure 4B, IA1), but in the inner cortex (Figure 4B, IA2) and the core (Figure 4B, IA3), AQP5 labels the sutures and this line of labelling in the inner cortex is substantially reduced in the lenses treated with pilocarpine (Figure 4C, IA2). The pre-incubation of the lenses with the TRPV1 inhibitor A-88 in the absence of pilocarpine has no effect on the membrane localisation of AQP5 in the inner cortex (Figure 4D, IA2), and the subsequent addition of pilocarpine in the continual presence of A88 (Figure 4E, IA2) fails to induce the removal of AQP5 observed with the addition of pilocarpine alone (Figure 4C, IA2).

### 2.3. Effect of Direct Activation of TRPV1 on Surface Pressure and Subcellular Localisation of AQP5 in the Rat Lens

Having established that pilocarpine-induced reduction in zonular tension increases both lens surface pressures, and that the removal of AQP5 from the tips of fiber cells that form the sutures in the inner cortical region of the anterior influx zone is blocked by inhibiting TRPV1 with A-88, we next investigated whether the direct activation of TRPV1 can mimic these effects of pilocarpine. As we found previously for the mouse [22] and bovine [21] lenses, the application of TRPV1 and TRPV4 activators also resulted in reciprocal increases and decreases in surface pressure in the rat lens (Figure 5). Application of the TRPV1 activator capsaicin resulted in a transient increase in lens surface pressure (Figure 5A) that reached a maximum (T_Max_) of 11.87 ± 0.96 mmHg at around 50 min after the start (T_0_) of drug administration. From this maximum value, there was then a gradual recovery of pressure over time towards the initial baseline (T_120_ = 1.66 ± 0.07 mmHg) at round 120 min post-drug application (Figure 5B). In contrast, the application of the TRPV4 activator GSK caused a transient decrease in lens surface pressure (Figure 5C) that reached a peak of −24.12 ± 2.20 mmHg at around 30 min after the start (T_0_) of drug administration. This initial significant reduction in lens pressure was followed by a gradual return of the pressure towards the baseline, which stabilised at a level of −2.31 ± 1.72 mmHg at 120 min post-drug application (Figure 5D). This observed biphasic increase and decrease in the rat surface hydrostatic pressure suggested that as observed for the mouse [19,32] and bovine [21] lenses, TRPV1 and TRPV4 channels in the rat lens also act as part of a dual feedback system that functions to main lens pressure constant. To further confirm that the return of surface pressure towards baseline following the activation of TRPV1 is indeed mediated by subsequent activation of TRPV4 channels, the lenses were first pre-incubated in the TRPV4 inhibitor HC-06 before activating TRPV1 via the application of capsaicin (Figure 5E). While blocking TRPV4 had no initial effect on surface pressure, the addition of capsaicin in the presence of HC-06 inhibited the biphasic response to capsaicin to produce a significant and sustained increase (T_120_ = 13.25 ± 0.94 mmHg) in lens surface pressure (Figure 5F). Thus, the sustained increase in surface pressure elicited by the combination of HC-06 and capsaicin appears to mimic the response obtained through the use of pilocarpine to increase ciliary muscle contraction and reduce the tension applied to the lens via the zonules.

Next, we investigated whether pharmacological modulation of the TRPV1 arm of this dual feedback pathway can also modulate membrane trafficking of AQP5 in the anterior influx (Figure 6) and equatorial efflux (Figure 7) pathways. Compared to the lenses incubated in the absence of capsaicin (Figure 6B), incubating the lenses in the presence of the TRPV1 activator for 60 min did not alter the localisation of AQP5 to the tips of fiber cells that form the anterior suture in the inner cortex of the rat lens (Figure 6C, IA2). Similarly, incubating the lenses in the TRPV4 inhibitor, HC-06, had no effect on the localisation of AQP5 (Figure 6D, IA2). However, pre-incubation of the lenses in the TRPV4 inhibitor for 30 min, followed by the addition of the TRPV1 activator capsaicin in the continued presence of HC-06 for 45 min, produced a sustained increase in surface pressure and induced the localised removal of AQP5 from the fiber cell tips that form the anterior sutures in the inner cortical region of the rat lens (Figure 6E, IA2), which mimicked the observations following the application of pilocarpine (Figure 4C, IA2).

The pharmacological modulation of the TRPV1 arm of the dual feedback pathway to mimic the effect pilocarpine on lens surface pressure also altered the membrane trafficking of AQP5 in peripheral fiber cells located in the efflux zone of the rat lens (Figure 7). Similar to what we had previously shown [31], pilocarpine application resulted in the removal of AQP5 from the lateral membrane domains of differentiating fiber cells at the lens equator to produce a cytoplasmic pattern of AQP5 labelling (Figure 7C), when compared to control lenses (Figure 7B). This effect of pilocarpine could be inhibited by pre-incubation of the lenses in the TRPV1 channel inhibitor A-88 (Figure 7D). While the direct addition of individual TRPV1 and TRPV4 modulators had no effect on AQP5 membrane location in the absence of pilocarpine, the sequential addition of HC-06 and capsaicin, which was used to produce a sustained increase in surface pressure that mimicked the actions of pilocarpine, resulted in the removal of AQP5 from the lateral membrane domains and the appearance of a cytoplasmic-labelling pattern (Figure 7E).

Taken together, our data show that changes to the tension applied to the rat lens via the zonules activates a TRPV1-mediated signalling pathway that produces sustained increase in lens surface pressure and spatially distinct changes to the membrane localisation of AQP5 in the anterior influx and equatorial efflux zones.

## 3. Discussion

In previous studies, we showed that changes to zonular tension induced by the addition of pilocarpine to the eye increased lens hydrostatic pressure in the mouse lens [22] and removed AQP5 from the membranes of fiber cells in the anterior influx and equatorial efflux zones of the rat lens [31]. In this current study, we chose the rat lens as our animal model to determine whether these observed effects of pilocarpine on AQP5 membrane trafficking are mediated by the activation of TRPV1 channels that have been shown to regulate lens hydrostatic pressure in the mouse lens [19,22]. Using a microelectrode-based pressure measurement system, we showed, for the first time in the rat lens, that surface pressure increased due to a decrease in zonular tension caused by the contraction of the ciliary muscle induced by pilocarpine (Figure 2A), and that this increase in pressure was abolished in the presence of the TRPV1 inhibitor A-88 (Figure 2B). Furthermore, we showed that the ability of pilocarpine to produce sustained increase in surface pressure in the rat lens could be mimicked by the activation of TRPV1 by capsaicin, but only if the lenses were pre-treated with the TRPV4 inhibitor HC-06 (Figure 5E), since the activation of TRPV1 in the absence of HC-06 caused a biphasic change in lens surface pressure (Figure 5A). Subsequent immunohistochemical experiments showed that the effects of pilocarpine on the removal of AQP5 from the membrane in the inner cortex of the anterior influx pathway (Figure 4E) and equatorial efflux zone (Figure 7D) could be inhibited by the addition of the TRPV1 inhibitor A-88. Furthermore, the addition of the combination of HC-06 + capsaicin to inhibit TRPV4 and activate TRPV1, respectively, to mimic the effects of pilocarpine on surface pressure also caused AQP5 to be removed from the membranes of fiber cells in the inner cortex of the anterior influx pathway (Figure 6E) and equatorial influx zone (Figure 7E). Taken together, these results show that decreasing the tension applied to the lens increases hydrostatic pressure and causes the removal of AQP5 from the membranes of fiber cells in spatially distinct areas of the lens via a mechanism that utilises the mechanosensitive channel TRPV1 to sense changes in zonular tension. The significance of this dual regulation via the changes in the tension applied to the lens on bulk water transport (lens pressure) and local water permeability (AQP5 membrane trafficking) on the overall function of the lens will be discussed.

Since in other tissues [29], including lens fiber cells [30], trafficking of AQP5 to the membrane is associated with an increase in P_H2O_, our data would suggest that the pilocarpine-induced removal of AQP5 will specifically reduce P_H2O_ in the anterior influx and equatorial efflux zones. Since this putative pilocarpine-induced reduction in P_H2O_ can be inhibited by blocking TRPV1 (Figure 4E and Figure 7D), or stimulated by the dual actions of activating TRPV1 and blocking TRPV4 (Figure 6E and Figure 7E), it suggests that AQP5-mediated changes to fiber cell P_H2O_ need to be included as a component of the TRPV1-mediated arm of the dual feedback system (Figure 8), which regulates the lens hydrostatic pressure gradient. In this revised model, activation of TRPV1 not only induces a rapid increase in ion uptake by NKCC1 to increase cellular osmolarity [20], but also a removal of AQP5 channels from the membrane that decreases P_H2O_. At this stage, we have only incorporated a change in AQP5-mediated P_H2O_ into the TRPV1-mediated arm of the dual feedback loop as we have no data to suggest that TRPV4 increases P_H2O_. This is simply a limitation of our immunolabelling experiments where we currently have no means to quantify an increase in AQP5 membrane labelling induced by an increase in TRPV4 activity, but it can clearly resolve the removal of AQP5 from the membrane induced by pilocarpine-induced activation of TRPV1.

Our potential identification of a change in AQP5-mediated P_H2O_ in the rat lens provides a potential answer to the AQP0 puzzle first posed by Hall and Mathias [33]. These authors raised the issue of the physiological relevance of the extensive regulation of AQP0-mediated P_H2O_ observed in vitro [34,35] to overall water transport in the lens observed in vivo [17,19]. These authors argued on the basis that fiber cell P_H2O_ is very large relative to sodium permeability, altering P_H2O_ of fiber cell membranes by a factor of 2–4 as observed in in vitro studies would have essentially no effect on fluid flow and would produce only a very tiny change in the intracellular sodium concentration [33]. They argued this is because water flow entering fiber cell membranes is essentially isotonic. In the lens, Na^+^ leak permeability is quite small, so P_H2O_ only needs to be small to allow fluid flow driven by very small sodium concentration gradients, essentially at the isotonic limit [36]. This then raises the question regarding why P_H2O_, especially its regulation, matters. In support of the argument that AQP0 P_H2O_ does not matter, the authors showed that a reduction in P_H20_ of fiber cell membranes in transgenic lens (AQP0^−/+^) had no effect on the pressure gradient profile, suggesting no effect on the regulation of water flow [33]. However, the authors conceded in their commentary that they had not considered a role for AQP5 in the regulation of water flow. Hence, our discovery of a potential role for AQP5 membrane trafficking in the regulation of lens pressure provides a potential answer to this puzzle, which can be confirmed by studying the role of AQP5 in lens pressure regulation in AQP5 WT and knockout lenses.

Based on the distinct spatial localisation of AQP5 membrane trafficking events at the apical tips of fiber cells in the inner cortex of the anterior sutural influx pathway (Figure 4 and Figure 6) and the lateral membranes of peripheral fiber cells in the equatorial efflux zone (Figure 7), we would predict that any effects on overall lens water transport may be subtle or localised to these areas. Removal of AQP-mediated P_H2O_ from the apical tips of fiber cells would be expected to reduce the movement of water from the extracellular space into fiber cells located in the inner cortex of the lens and could affect the volume of the cells in this region. In the human lens, it has been shown that it is the anterior curvature that undergoes the largest changes in shape in response to changes in zonular tension that occur during accommodation [37,38,39]. In the non-accommodating rat lens, it is our working hypothesis that changes in zonular tension cause localised changes in the water content specifically in the anterior region of the lens, which may alter the anterior surface curvature of the lens and, hence, the steady state optical power of the lens. In contrast, a decrease in P_H2O_ induced by the removal of AQP5 from fiber cell membranes in the equatorial efflux zone would reduce the removal of water from the lens. This decrease in P_H2O_ in the equatorial efflux zone contributes to both the increase in surface pressure observed in the present study (Figure 2), the overall hydrostatic pressure gradient observed in the mouse lens [22], and, therefore, to the overall regulation of lens power [1].

In summary, we have linked the membrane trafficking of AQP5 induced by changes in the zonular tension applied to the lens to the regulation of lens hydrostatic pressure via the activation of the mechanosensitive channel TRPV1. By including the spatial modulaton of P_H2O_ in the anterior influx pathway and the equatorial efflux zone, we have advanced new working hypotheses to explain how the lens actively regulates its optical power. Testing these hypotheses using a combination of immunolabelling [28], pressure measurements [22], and in vivo MRI to measure lens water content, geometry (volume), the water-to-protein ratio (GRIN), and lens power [40] on wild-type and AQP5 knock out lenses will be the focus for future work.

## 4. Methods and Materials

### 4.1. Reagents and Buffers

All chemicals were of analytical grade and, unless otherwise stated, were purchased from Sigma Aldrich (St. Louis, MO, USA). The lenses were incubated in an Artificial Aqueous Humour (AAH; 125 mM of NaCl, 4.5 mM of KCl, 0.5 mM of MgCl_2_, 10 mM of NaHCO_3_, 2 mM of CaCl_2_, 5 mM of glucose, 10 mM of sucrose, 10 mM of HEPES, pH of 7.4, 300 mOsmol/L). Phosphate-buffered saline (PBS) was prepared from tablets containing 137 mM of NaCl, 2.7 mM of KCl, and 10 mM of phosphate buffer at a pH of 7.4. Blocking buffer for immunolabeling was prepared from 3% bovine serum albumin and 3% normal goat serum diluted in PBS with a pH of 7.4. Additionally, 10%, 20% and 30% sucrose solutions for cryoprotection were prepared in PBS with a pH of 7.4. Pilocarpine (0.2%) was used to contract the ciliary muscle [41]. The TRPV4 agonist GSK-1016790A (Abcam, Cambridge, MA, USA) was used at concentration of 30 nM [42,43], while capsaicin (Abcam, Cambridge, MA, USA) was used at a concentration of 10 μM to activate TRPV1 channels [44,45]. HC-067047 (Abcam, Cambridge, MA, USA) and A-889425 (Abcam, Cambridge, MA, USA) were both used at a final concentration of 10 μM to inhibit TRPV4 [46] and TRPV1 channels [47], respectively. The rabbit anti-AQP5 C-terminus antibody (Merck Millipore, Darmstadt, Germany) was used at 1:100 dilution for immunolabeling. The secondary antibody goat anti-rabbit IgG AlexaFluor 488 conjugate was used at 1:100 dilution and purchased from Life Technologies (Carlsbad, CA, USA). In the control experiments, we utilised only the secondary antibody with no background observed. The wheat germ agglutinin (WGA) Alexa Fluor 594 conjugate was used as a membrane marker at 1:100 dilution and purchased from Thermo Fisher Scientific (Waltham, MA, USA). 4′,6-Diamidine-2′-phenylindole dihydrochloride (DAPI)was used as a cell nucleus marker; it was prepared as 1.25 mg/mL stock and diluted to 1:100 final concentration.

### 4.2. Lens Preparation

All animal experiments were carried out in accordance with the ARVO Statement for the Use of Animals in Ophthalmic and Vision Research and approved by the University of Auckland Animal Ethics Committee, protocol code AEC1413. P21–28-old *Wistar rats* of either sex were supplied by the Vernon Jansen Unit (VJU), located in the Faculty of Medical and Health Sciences at the University of Auckland. The rats were euthanised via CO_2_ asphyxiation, and their eyeballs were rapidly removed using a pair of curved surgical scissors and transferred to a Petri dish containing AAH that was pre-warmed to 37 °C. The eyes were dissected under a binocular dissecting microscope (Nikon, Tokyo, Japan) using micro spring scissors (World Precision Instruments, Sarasota, FL, USA) in two different ways, and subsequently used to measure lens hydrostatic pressure measurements or AQP5 membrane trafficking. For the lens pressure measurements, four cuts radiating outwards from the optic nerve towards the limbus were made to create four flaps from the sclera that were subsequently pinned to the bottom of a recording chamber. The bottom of the recording chamber was coated with SYLGARD™ 184 (Dow Chemical Corporation, Midland, MI, USA) and had a central pit into which the dissected eye was positioned to ensure that the pinned eye and the lens did not move during microelectrode impalement. This preparation left the lens connected to the ciliary body via the lens zonules while exposing the lens to enable the insertion of the microelectrode. In this configuration, different combinations of pharmacological modulators could be added directly to the bath. To assess the pharmacological regulation of AQP5 membrane trafficking, a small window of ~2 mm was cut into the cornea of the enucleated eye to facilitate the penetration of pharmacological modulators into the eye while ensuring that the tension applied to the lens via the zonules was maintained (in vivo preparation).

### 4.3. Measurements of Lens Intracellular Hydrostatic Pressure

Lens intracellular hydrostatic pressure was measured using an approach based on the manometer-microelectrode method described by Gao et al. [17], in which the manometer was replaced by a picospritzer [21]. Briefly, a voltage microelectrode having a resistance of 1.5 to 2.5 MΩ was back filled with 3M of KCl, inserted into a holder, and then mounted on a manual micromanipulator. The microelectrode tip resistance (R_e_) was first measured in the bathing solution outside of the lens by passing current pulses through the microelectrode using TEV-200 amplifier (Dagan Corporation, Minneapolis, MN, USA) and recording the voltage response. The microelectrode was then advanced at a 45° angle through the lens capsule to impale lens fiber cells near the surface of the lens. Once in the lens, the positive intracellular pressure pushed fiber cell cytoplasm into the microelectrode tip, causing an increase in the tip resistance (ΔR). The pressure within the microelectrode was adjusted using a PLI-100 picospritzer (Harvard Apparatus, Holliston, MA, USA) connected to the side port of the microelectrode holder until the cytoplasm was pushed out of the tip, and the resistance returned to its original value (see Figure 1 [21]). This applied pressure was then taken as the lens intracellular hydrostatic pressure at that specific location in the lens.

To study how the pressure in lens surface cells is regulated, the position of the microelectrode was held constant, and changes in hydrostatic pressure (ΔP) at the lens surface, in response to the application of pharmacological modulators, were recorded over time. Surface pressure was monitored for a period of 30 min to establish a control baseline pressure before the addition of pharmacological modulators. Following the administration of pharmacological reagents, the pressure at that same location was monitored and recorded for an additional 120 min. The effect of the application of a pharmacological reagent on the surface hydrostatic pressure was expressed as the change in hydrostatic pressure (ΔP) by subtracting the initial baseline surface pressure measured in the absence of the pharmacological reagent and plotted as a function of time. The subtraction of the initial pressure from the trace of the response allowed us to account for the uncontrolled differences in the depth of penetration of the microelectrode and to normalize the data so that we were able to compare the data from different lens experiments. Finally, the time course of the mean change of the response (±SEM) of surface pressure was calculated, which allowed us to examine the response using one response curve. The number of lenses used to measure the surface hydrostatic pressure with and without pharmacological reagents was at least 4.

### 4.4. Lens Organ Culture, Fixation, and Immunolabelling

To investigate the mechanisms regulating AQP5 trafficking in vivo, the lens preparations were organ cultured in a Heracell 150i CO_2_ incubator (ThermoFisher Scientific, Waltham, MA, USA) in AAH for up to 60 min with or without various combinations of TRPV1 and TRPV4 channel modulators and/or pilocarpine to change the tension applied to the lens via the zonules (see results for details). The experiments were terminated by adding freshly prepared 0.75% PFA diluted in PBS with a pH of 7.4 for 24 h at room temperature to fix the lens prior to it being processed for cryosection and immunolabelling using established techniques [48]. The cryoprotected lenses were positioned on chucks in an axial orientation, encased in Tissue-Tek O.C.T. compound (Sakura Finetek, Tokyo, Japan), and snap frozen for 15 s in liquid nitrogen. The axial sections were cut using a Leica CM3050S cryostat (Leica Biosystems, Heidelberg, Germany), and consecutive sections of ~14 µm in thickness were collected from at least three lenses for each experimental condition to ensure the consistency of immunolabelling results. The sections were washed three times for 5 min in PBS with a pH of 7.4. The lens sections were first treated with 0.01% Triton-X 100 diluted in PBS with a pH of 7.4 for permeabilization of cell membranes for 15 min and then treated with a blocking buffer containing 3% normal goat serum and 3% bovine serum albumin diluted in PBS with a pH of 7.4 for 1 h at room temperature to prevent the non-specific binding of antibodies to tissue or to Fc receptors. Anti-AQP5 primary antibody was diluted in the blocking buffer and applied to the sections for overnight incubation at 4 °C. After the overnight incubation, the sections were washed three times for 10 min in PBS with a pH of 7.4 to remove unbound primary antibody. Goat anti-rabbit Alexa Fluor 488 secondary antibody was diluted in the blocking buffer and then applied to the washed sections for 2 h at room temperature. Following three additional 10 min washes in PBS with a pH of 7.4 to remove unbound secondary antibody, the sections were incubated in WGA and DAPI diluted in PBS with a pH of 7.4 for another 2 h at room temperature and in the dark for fluorescent labelling of cell membranes and nuclei, respectively. The slides were then washed three times for 10 min in PBS with a pH of 7.4, and excess solution was removed before the application of an anti-fade mounting medium (VECTASHIELD HardSet, Vector Laboratories, Newark, CA, USA). Finally, the slides were mounted with a coverslip (Thermo Fisher Scientific, Waltham, MA, USA), sealed with nail polish, and then stored at 4 °C in the dark until being imaged using confocal microscopy (Olympus FV 1000, Tokyo, Japan). The acquired images from separate channels were processed using the Adobe Photoshop software (Adobe Systems Incorporated, San Jose, CA, USA).

### 4.5. Statistical Analysis

To evaluate the surface hydrostatic pressure of the control lenses and the lenses treated with TRPV1 and TRPV4 modulators and/or pilocarpine, data were obtained from separate experiments and contained at least four lenses per condition, while immunolabelling data were obtained from three separate lenses. The data of the experiments are presented as mean ± standard error of the mean (SEM). The graphs were plotted using the GraphPad Prism 8 software (La Jolla, CA, USA). The statistical significance of the data was tested based on the non-parametric Mann–Whitney U test using a statistical software (SPSS Statistics Data Editor Version 27, Chicago, IL, USA). Statistical significance was set at the α = 0.05 level.

## Figures and Tables

**Figure 2 ijms-24-09080-f002:**
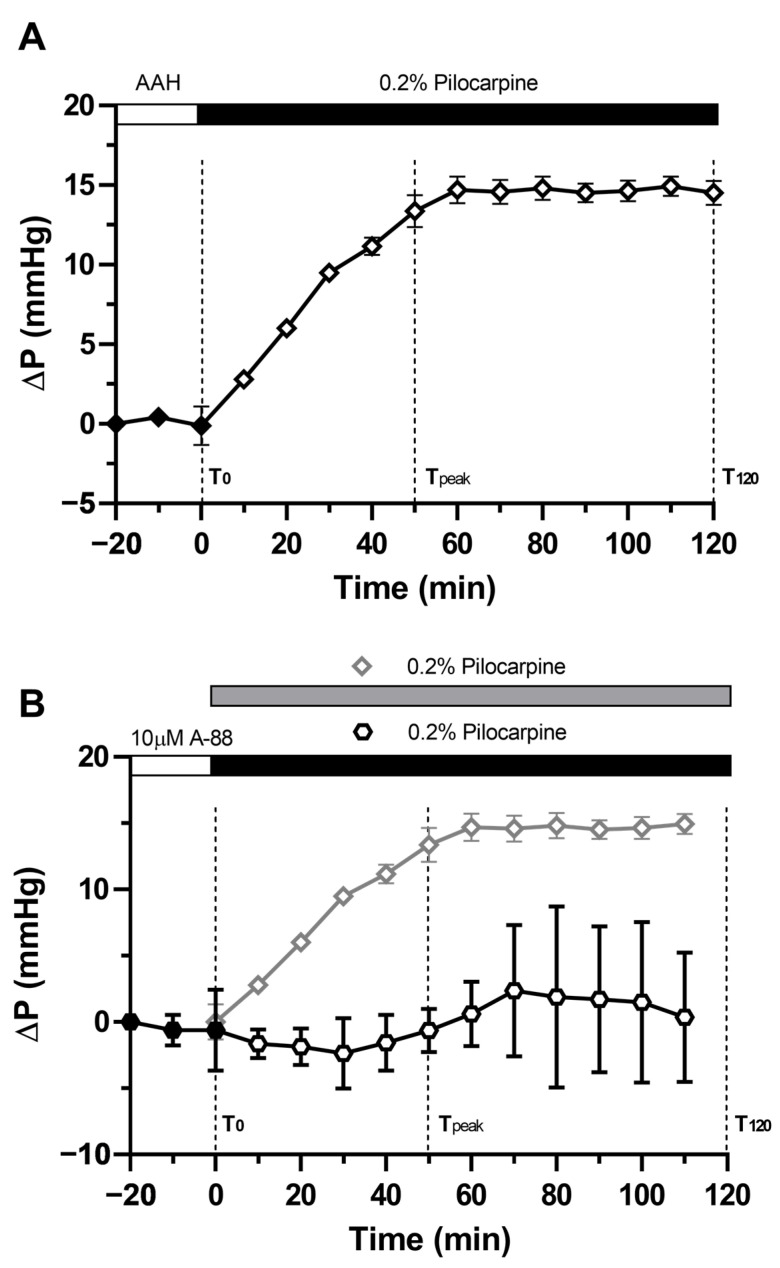
Effect of pilocarpine on the surface pressure of the rat lens. (**A**) Time course of the average change in surface pressure (ΔP) in response to the application of 0.2% pilocarpine (n = 6). (**B**) Time course of the average ∆P in response to the application of 0.2% pilocarpine in the presence of 10 µM of A-88 (n = 6). For comparison, the sustained increase in pressure observed in the lenses treated with pilocarpine is only overlaid on the graph.

**Figure 3 ijms-24-09080-f003:**
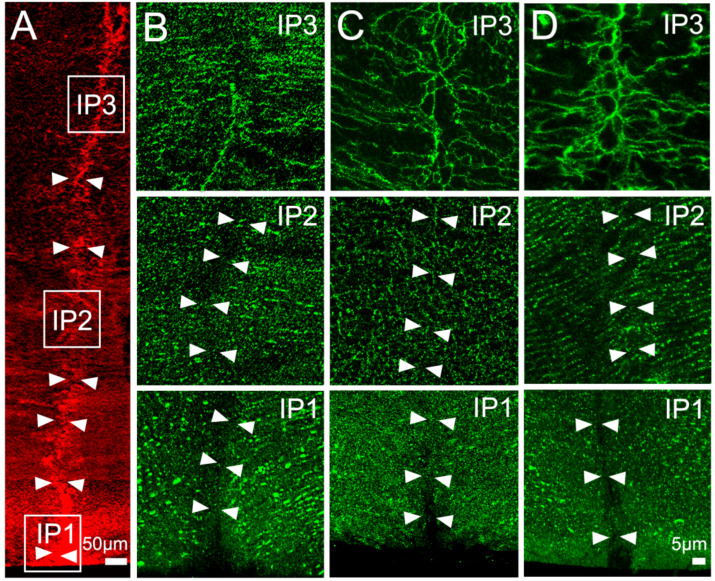
Effect of pilocarpine on the subcellular localisation of AQP5 in the posterior influx zone of the rat lens. (**A**) Representative image montage of a posterior suture spanning from the outer cortex to the core of the lens that is labelled with the membrane marker WGA (red). The white boxes represent the regions within the sutures from where the high-magnification images were captured to show the localisation of AQP5 protein (green). The arrows are used to indicate the location of the sutures. (**B**) In the control lenses, AQP5 is found missing from the basal tips of fiber cells that form the posterior sutures in the IP1 and IP2 regions, but AQP5 does not colocalise within the suture in the lens core (IP3) region. (**C**,**D**) The addition of pilocarpine alone (**C**) or after the preincubation of the lenses in the TRPV1 inhibitor A-88 (**D**) causes no change to the subcellular distribution of AQP5 along the length of the posterior suture.

**Figure 4 ijms-24-09080-f004:**
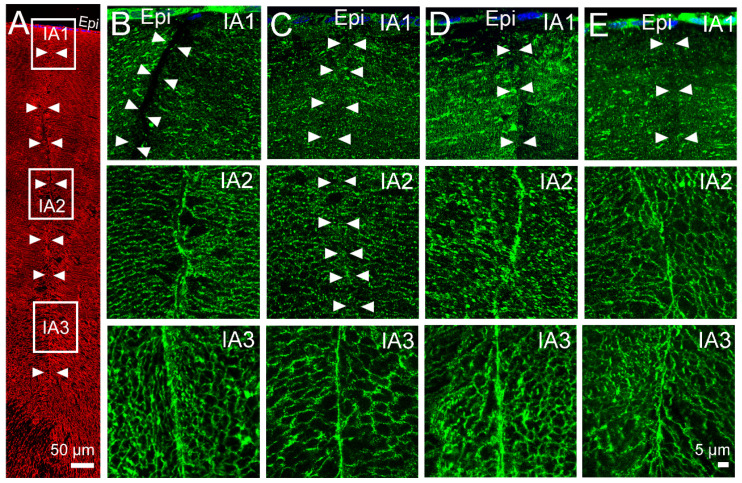
Effect of pilocarpine on the subcellular localisation of AQP5 in the anterior influx zone of the rat lens. (**A**) Image montage of the anterior suture spanning from the outer cortex to the core of the lens that is labelled with the membrane marker WGA (red). The white boxes represent the regions within the sutures from where high-magnification images were captured to show the localisation of AQP5 protein (green). The arrowheads are used to indicate the location of the sutures. (**B**) In the control lenses, AQP5 is found missing from the apical tips of fiber cells that form the anterior suture in the outer cortical (IA1) region, but it is associated with the sutures in the inner cortex (IA2) and core (IA3) regions of the lens. (**C**) In the presence of pilocarpine, AQP5 remains absent from the suture in the outer cortex (IA1), but it is missing from the sutures in the inner cortex (IA2) and remains present in the core (IA3). (**D**) Pre-incubation of the lens in the TRPV1-inhibitor HC-06 causes no change to the subcellular distribution of AQP5 along the length of the anterior suture. (**E**) Pre-incubation of the lens in HC-06 for 30 min, followed by the addition of pilocarpine, inhibits the removal of AQP5 from the anterior suture in the inner cortical region (IA2) of the rat lens. Epi—epithelium.

**Figure 5 ijms-24-09080-f005:**
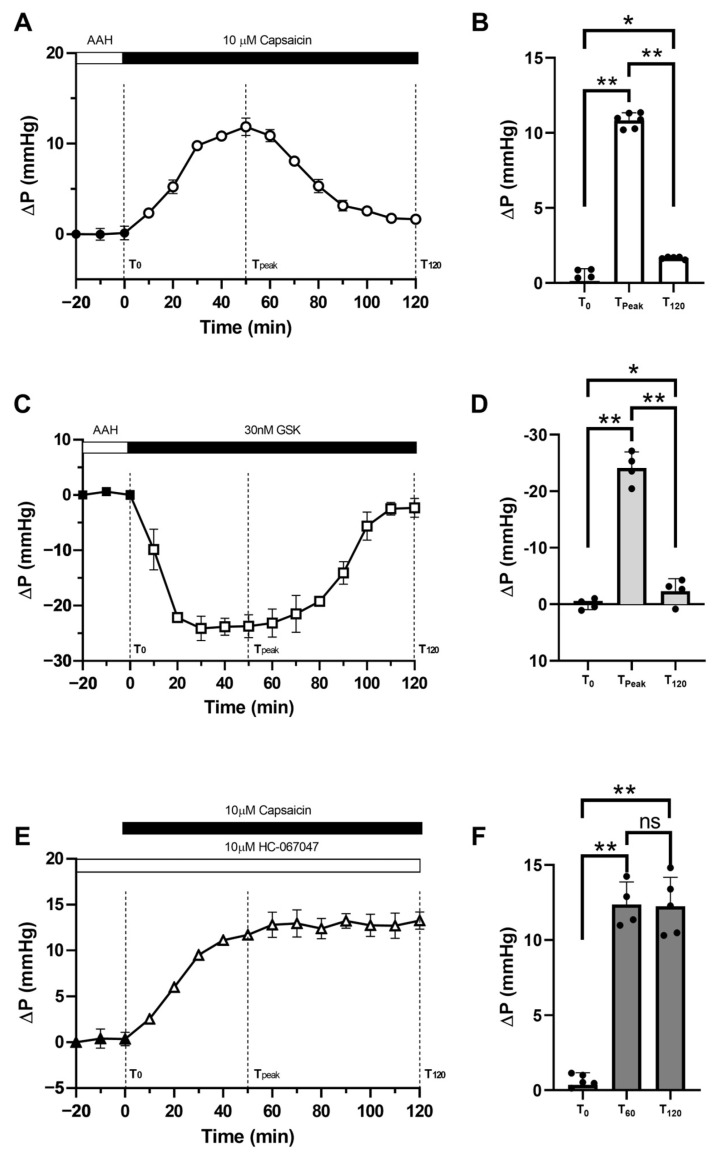
Effects of pharmacological modulation of TRPV1 and TRPV4 activities on surface pressure in the rat surface lens. (**A**) Time course of the biphasic increase in average in surface pressure (ΔP) in response to the application of 10 µM of capsaicin (n = 6). (**B**) Statistical comparison of average ∆P at the start of the experiment (T_0_), at the peak of the response (T_Max_), and at 120 min (T_120_) post capsaicin application. (**C**) Time course of the biphasic decrease in ∆P in response to the application of 30 nM of GSK (n = 4). (**D**) Statistical comparison of ∆P at the time points T_0,_ T_Max_, and T_120_ post GSK application. (**E**) Time course of the sustained increase in ∆P in response to the application of 10 µM of capsaicin in the presence of 10 µM of HC-06 (n = 6). (**F**) Statistical comparison of ∆P at the time points T_0_, T_Max_, and T_120_ post HC-07 + capsaicin application. All data are presented as mean ± SEM. The differences in the lens pressure response between the treatment groups at each time point were assessed by conducting independent-sample Mann–Whitney *U* Test. * (*p* < 0.05) and ** (*p* < 0.01) indicate a significant difference between the compared values. ns—not statistically significant.

**Figure 6 ijms-24-09080-f006:**
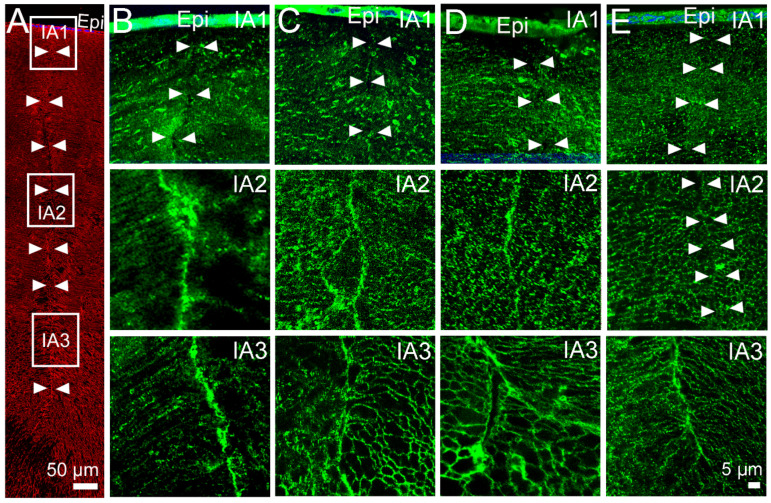
Effects of pharmacological modulation of TRPV1 and TRPV4 activities on subcellular localisation of AQP5 in the anterior influx region of the rat lens. (**A**) Image montage of the anterior suture spanning from the outer cortex to the core of the lens that is labelled with the membrane marker WGA (red). The white boxes represent the regions within the sutures from where high-magnification images were captured to show the localisation of AQP5 protein (green). The arrows are used to indicate the location of the sutures. (**B**). In the control lenses incubated in AAH, AQP5 is found missing from the apical tips of fiber cells that form the anterior suture in the outer cortex (IA1), but it is present in the sutures of the inner cortex (IA2) and the core (IA3) of the lens. (**C**,**D**) In the presence of the TRPV1 activator capsaicin (**C**) or the TRPV4 inhibitor HC-06 (**D**), AQP5 remains absent from the suture in the outer cortex (IA1) and present in the sutures of the inner cortex (IA2) and core (IA3) regions. (**E**) The pre-incubation of the lenses in HC-06 for 30 min, followed by the addition of capsaicin for another 45 min, results in the removal of AQP5 from the suture in the inner cortex (IA2), but it has no effect on AQP5 labelling in the outer cortex (IA1) and core (IA3) regions. Epi—Epithelium.

**Figure 7 ijms-24-09080-f007:**
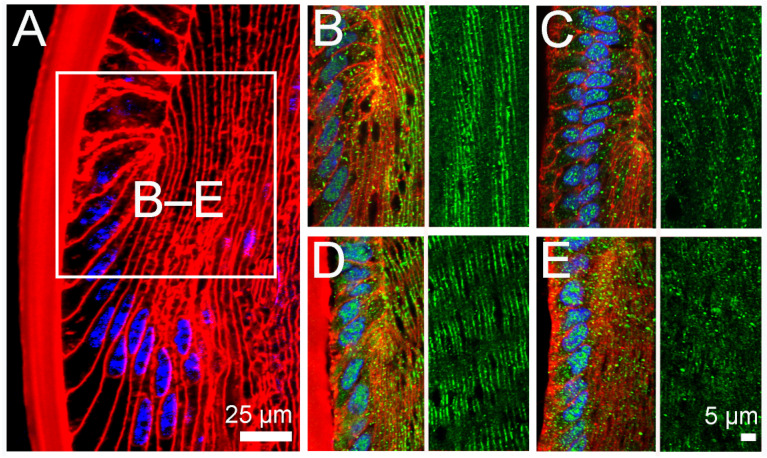
Effects of pharmacological modulation of TRPV1 and TRPV4 activities on the subcellular localisation of AQP5 in the equatorial efflux region of the rat lens. (**A**) Image of the lens equator labelled with the membrane (WGA, red) and nucleus (DAPI, blue) markers show the zone of transition where equatorial epithelial cells differentiate into fiber cells and undergo extensive elongation of their lateral membranes. At this stage of differentiation, the apical and basal tips of fiber cells are associated with the epithelium and capsule, respectively. The white box represents the region from where high-magnification images (**B**–**E**) were captured to show the localisation of AQP5 protein (green). (**B**) In the lenses incubated in AAH for 60 min, AQP5 is found localised to the lateral membranes of differentiating fiber cells of the outer cortex. (**C**) In the lenses treated with pilocarpine for 60 min, the AQP5 distribution shifts to a cytoplasm-labelling pattern. (**D**) Addition of pilocarpine to the lenses pre-incubated in the TRPV1 inhibitor A-88 for 30 min maintains the association of AQP5 with the membrane in this zone of the lenses. (**E**) Exposure of the lenses to the TRPV4-inhibitor HC-06 for 30 min, followed by the addition of the TRPV1-activator capsaicin to mimic the effects of pilocarpine on lens pressure, causes a shift in AQP5 from the membrane to the cytoplasm. To reveal the membrane and cytoplasmic localisation of AQP5, the magnified images (**B**–**E**) are presented as split images that include single AQP5 areas and adjacent areas labelled with both AQP5 and WGA.

**Figure 8 ijms-24-09080-f008:**
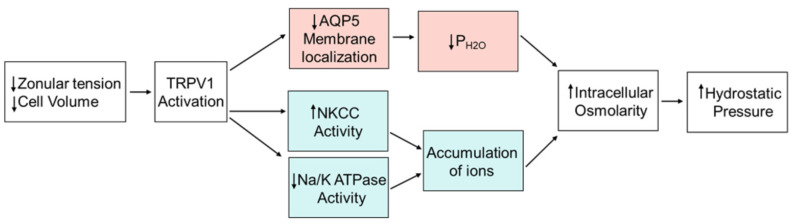
Revised model of regulation of lens hydrostatic pressure by zonular tension. A simplifed model showing only the TRPV1-mediated arm of the dual feedback system that regulates hydrostatic pressure in the lens. In this model, pilocarpine-induced decrease in the zonular tension applied to the lens activates a TRPV1-mediated signalling pathway that immediately increases the activity of NKCC to increase the uptake of ions and, after a delay, reduces the activity of NaKATPase to reduce the removal of ions from surface fiber cells and affect the intracellular accumulation of ions. In parallel to these changes in ion transporter activity, the removal of AQP5 from the membrane in the anterior influx pathway and equatorial efflux zone by TRPV1 activation specifically reduces P_H2O_ in these regions of the lens. These changes in transporter activity and P_H2O_ combine to increase the intracellular osmolartity and, in turn, the hydrostatic pressure gradient. Futhermore, based on the observed ability of hyperosmotic challenge to increase NKCC1 activity via a TRPV1-mediated signalling pathway [20], we would predict that hyperosmotic challenge would also modulate AQP5 membrane trafficking.

## Data Availability

The data presented in this study is available in this article.

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
