# Peer review of "Modulation of Membrane Trafficking of AQP5 in the Lens in Response to Changes in Zonular Tension Is Mediated by the Mechanosensitive Channel TRPV1"

_ijms, 2023, doi:10.3390/ijms24109080_

Round 1

Reviewer 1 Report

In this manuscript, the authors mainly report similar findings occurred in the rat lenses, like previously published results in the mouse lenses, about the zonule activates a TRPV1-mediated signaling pathway to produce a sustained increase in lens surface pressure and spatially distinct changes to the membrane localization of AQP5 in the anterior and equatorial regions of lens fibers. Overall quality of data is acceptable. The conclusions are sound and the data confirm that the similar mechanism in both rat and mouse lenses about the modulation of the movement of lens water, which may depend on the membrane localization of AQP5 to regulate PH2O and hydrostatic pressure. There are a few minor concerns about the data interpretations are listed below.

Figure 4C, based on the signals in IA2 panel, in presence of pilocarpine, AQP5 was obviously reduced rather than “missing” from the sutures in inner cortex (IA2). The suture should be clearly indicated in the panel.

Figure 6 legend, AQP5 was found missing from the “apical tips” of fiber cells that form… “apical tips” should be changed as “anterior tips” because the ends of these inner fibers in the suture are no longer detached to apical sides of epithelium.

It’s difficult to see the suture in Figure6D-IA3 panel (with HC-06 treatment). Perhaps, the suture in these panels should be clearly indicated to ensure whether or not the distribution of AQP5 remains unchanged at the ends of fibers.

Author Response

In this manuscript, the authors mainly report similar findings occurred in the rat lenses, like previously published results in the mouse lenses, about the zonule activates a TRPV1-mediated signaling pathway to produce a sustained increase in lens surface pressure and spatially distinct changes to the membrane localization of AQP5 in the anterior and equatorial regions of lens fibers. Overall quality of data is acceptable. The conclusions are sound and the data confirm that the similar mechanism in both rat and mouse lenses about the modulation of the movement of lens water, which may depend on the membrane localization of AQP5 to regulate PH2O and hydrostatic pressure. There are a few minor concerns about the data interpretations are listed below.

We would like to thank the referee for their critical review of our manuscript.

Figure 4C, based on the signals in IA2 panel, in presence of pilocarpine, AQP5 was obviously reduced rather than “missing” from the sutures in inner cortex (IA2). The suture should be clearly indicated in the panel.

R: We have changed the text to “AQP5 was reduced” instead of missing and have used arrow heads to indicate the sutures in Figure 4C IA2.

Figure 6 legend, AQP5 was found missing from the “apical tips” of fiber cells that form… “apical tips” should be changed as “anterior tips” because the ends of these inner fibers in the suture are no longer detached to apical sides of epithelium.

R: Our use of the term apical tip does not only refer to the association of the fiber cells with the apical surface of the anterior epithelium. It also refers to the polarity of the fiber cells that maintain their apical and basal membrane domains as their lateral membrane undergo extensive elongation (see Figure 1C). Hence, we have retained the reference to apical tips of the fiber cells forming the anterior suture.

It’s difficult to see the suture in Figure6D-IA3 panel (with HC-06 treatment). Perhaps, the suture in these panels should be clearly indicated to ensure whether or not the distribution of AQP5 remains unchanged at the ends of fibers.

R: Rather than utilising arrow heads to highlight the suture we have instead replaced this image with a different image that more clear shows the suture line.

Reviewer 2 Report

A mature mammalian lens is composed of an anterior epithelium that overly a core of elongated organelle-devoid cells (so called ‘fiber cells’) that continuously grow via the differentiation and maturation of epithelial to fiber cells initiated somewhere along the lens equator. Due to the largely avascular nature of lenses (due to vasculature thinning during development) the lens has co-opted dynamic systems to regulate oxygen, ion, and fluid availability to maintain the overall structure and functional maintenance of lens components. The lens has sub-tissue regions including the regions near the apical and basal fiber cell tips, zonular penetration from the ciliary, and sutures that vary in structure between mammalian species. This makes the lens an ideal model system to understand autocrine and paracrine regulation of tissue microcirculation.

In this manuscript entitled “Modulation of the membrane trafficking of AQP5 in the lens in response to changes in zonular tension is mediated by the mechanosensitive channel TRPV1” by Rosica S. Petrova and others, the combined group explores lens microcirculation in the light of zonular tension and mechanosensitive TRPV1 channel regulation of AQP5 lens tissue localization in non-accommodating rat lenses.

The authors utilize an elegantly robust lens organ culture system using artificial aqueous humor and with this ex-vivo lenses the group treated them with pharmaceuticals targeting or modulating TRP channel biology to block or specific TRP receptors on rat lenses ex vivo and utilized intracellular hydrostatic pressure detection and confocal microscopy with some statistical analysis. Via these methods the authors found that decreased zonular tension and therefore cell volume leads to mechanosensitive TRPV1 activation, this likely leads to an increase in intracellular osmolarity via AQP5 cytoplasmic redistribution and subsequent increase in lens hydrostatic pressure. 

The work is seemingly very robust, has some direct clinical relevance (pilocarpine for presbyopia that should be expanded on) yet could be refined and sharpened for a well-polished manuscript that is easily interpreted and allows for others to build upon it. Most important is probably the quantification and statistical analysis of confocal with replicate n’s on the quantification shown. 

Points raised in my assessment of the manuscript. 

0. Were any notable AQP5 redistribution seen in the epithelium? Please provide some epithelia confocal images regardless. 

1.Some of the figures are pretty blurry, especially Fig 5. Please make these vector formatted or  higher resolution. The confocals are of high resolution.  

2. For all bar charts please indicate each data point with a unique dot per measurement (Fig 5 B, D, F. Just overlay these ontop of the current charts which is easy to do in prism or other graphical software such as R.

3. Fig 8 would greatly benefit by showing the biological changes on the lens diagram shown in the first figure. Ideally this would make it a sort of graphical abstract and could maybe replace or work together with the written model given. I think this would greatly enhance the communication and interpretation for the reader as some of the data and outcomes are a little tricky to interpret. 

4. I know it is slated for future work, but the current work would greatly benefit from some structural lens perspectives with the changes beyond pressure changes and confocal, MRI could be very nice here. Even just some basic light microscopy of the lenses on grids.. For instance, does the shape of the overall lens change? What about via TEM, are their cytoplasmic changes? Is it possible to model or measure those microfluidic changes more specifically? This could show large-scale volumetric changes that may be occurring with the pharmacological treatments. 

5. Please provide comment on the specificity of all drugs. Maybe I missed it, my apologies if so.

6. Please provide a couple examples of secondary alone staining for confocal. Could be supplementary, just nice to have.

7. Please provide some overlay examples of membrane staining and AQP5 labeling for better visualization of membrane to cytoplasm trafficking. 

8. Please provide some rough quantification of all confocals to help assure the reader that the changes observed are meaningful. This could be overall intensity redistribution changes.  It is incredibly difficult to decipher these confocal images for the reader from just looking at the images, even at first glance they are kind of confusing to be honest and I am well-trained in confocal across many cell and tissue types including lenses. Adding some additional labels and outlining structures with annotation will greatly help the interpretation of these figures to the readers. 

9. The authors need to expand on those comments of the limitations of the study and discuss how future studies can circumvent these limitations:

-The mechanism of how exactly AQP5 is moved from the membrane is still not known, just that TRPV1 is involved and probably plays a key role as an initiator of some sort.

-It still remains enigmatic if TRPV1 and AQP5 directly interact, or this is an indirect phenomena. Could there be additional mediators required? Please expand on this.

-The chemical modulators could have off-target and probably lens penetrance effects, so a genetic model could be very useful. Please expand on this and provide any drug penetrance data available.

-The effect the findings have on lens transparency or homeostasis is not determined. 

-Correlation of findings with non-rodent lenses. How do the findings relate to lens evolution. How do the findings relate to human lens biology?

-How do the data relate to patients receiving pilocarpine for presbyopia?

-The experiments were performed on young rats, what about aged rats? Is this an age-specific phenomena in rodents?

-How do the data go beyond the lens to understand TRPV1 and AQP biology? This could be interesting to add since pilocarpine is used as a treatment for dry mouth.

-What happens to the overall biology of the lenses beyond just AQP5 with the pharmacological TRP modulation? This could have been addressed by a high-throughput method… (Some theoretically related work: https://www.ncbi.nlm.nih.gov/pmc/articles/PMC8476594/ ). Based on this ref, it is likely that much more than AQP5 is altered!!!! --- This should be commented on, maybe include this reference if you want to, and explain on it. I think it’s entirely ok to say this is a limitation, reiterating that the AQP5 redistribution is a key effect since it happens so quickly., and that future work will capitalize on high-throughput methods to find the full range of TRP channel lens requirements.

-What if the lens was cultured posteriorly with artificial vitreous and anteriorly with artificial aqueous, would that change the findings? (not sure this is possible with today’s tech but interesting to speculate on..)

10. Please provide a diagrammatic cartoon of the microelectrode equipment and experimental setup to figure 2 to visually explain how the surface pressures was determined. This will be very helpful for other biologists in the TRP field. 

11. The authors should have validated AQP5 redistribution modulation with the various treatments via another protein method, such as western blot or co-ip mass spec from plasma membrane and cytoplasmic fractions of treated lenses. It would be incredibly nice to include this. To be honest, I would be wary of any singular method that shows a redistribution, it is best to orthogonally confirm with another type of related experiment.

12. It would be nice to include a raw data policy, such as available upon request. Up to you but this will become standard sooner than later.

Author Response

The work is seemingly very robust, has some direct clinical relevance (pilocarpine for presbyopia that should be expanded on) yet could be refined and sharpened for a well-polished manuscript that is easily interpreted and allows for others to build upon it. Most important is probably the quantification and statistical analysis of confocal with replicate n’s on the quantification shown. 

We thank the reviewer for their comprehensive and helpful review of our manuscript. We would like to raise the issue of quantification up front in our response to the reviewer’s questions. Our focus in this paper has been on regional changes in the membrane distribution of AQP5 in fiber cells membranes in different influx and efflux regions of the lens. The fiber cells have a very ordered cellular architecture that facilitates this analysis and allows us to easily visual changes in the subcellular distribution of AQP5 from an ordered association with the fiber cell membrane to a disordered association with punctate-like cytoplasmic vesicles. In specific regions of the anterior influx pathway and equatorial efflux zone this shift from membrane to cytoplasmic AQP5 labelling patterns in response to pilocarpine addition are so obvious we have not seen it necessary to quantify the extent of this all or nothing re-distribution.

Points raised in my assessment of the manuscript. 

  1. Were any notable AQP5 redistribution seen in the epithelium? Please provide some epithelia confocal images regardless. 

R: We have provided in Figs 4 and Fig 6 some representative images of the distribution of AQP5 in the epithelium and have added labelling to better designate its localization.  As pointed out we have not focussed specifically on AQP5 subcellular distribution in the epithelial cells as our main interest was on fiber cells in the different regions of the lens. Epithelial cells have a less ordered membrane structure relative to fiber cells and AQP5 is primarily cytoplasmic in its distribution and remains so after the application of pilocarpine. What would be interesting to study is what would happen to the distribution of AQP5 in the epithelium following application of tropicamide that we have shown increases the tension applied to the lens via the zonules. However, such studies on the epithelium would require development of methods to quantify the changes to AQP5 subcellular location as we could not rely on the simple all or nothing shift in AQP5 labelling seen in the highly ordered fiber cells.

1.Some of the figures are pretty blurry, especially Fig 5. Please make these vector formatted or  higher resolution. The confocals are of high resolution.  

R: We have checked the resolution of all images and where required increased it to meet the required specification for publication.

  1. 2. For all bar charts please indicate each data point with a unique dot per measurement (Fig 5 B, D, F. Just overlay these on top of the current charts which is easy to do in prism or other graphical software such as R.

R: We have changed the presentation of the data in Figure 5 to display the individual data points.

  1. 3. Fig 8 would greatly benefit by showing the biological changes on the lens diagram shown in the first figure. Ideally this would make it a sort of graphical abstract and could maybe replace or work together with the written model given. I think this would greatly enhance the communication and interpretation for the reader as some of the data and outcomes are a little tricky to interpret. 

R: Thank you for this suggestion. We have already used a graphical abstract as part of the publication process that we believe fulfils this role.

  1. I know it is slated for future work, but the current work would greatly benefit from some structural lens perspectives with the changes beyond pressure changes and confocal, MRI could be very nice here. Even just some basic light microscopy of the lenses on grids. For instance, does the shape of the overall lens change? What about via TEM, are their cytoplasmic changes? Is it possible to model or measure those microfluidic changes more specifically? This could show large-scale volumetric changes that may be occurring with the pharmacological treatments. 

R: We agree with the reviewer that these would be useful measurements to add to our publication, but they are not as easy to obtain as you may think since we have to preserve the relationship between the ciliary muscle, zonules and the lens to observe the effects of the addition of pilocarpine on the anterior surface curvature. So simply photographing lens on grids following the cutting of the zonules to remove the lens from the eye would not capture these changes in overall lens morphology. To address this, we are currently investigating in a separate study using in vivo MRI on wildtype and AQP5 KO mice to visual changes to lens geometry and water content in vivo following pilocarpine addition.

Yes, we agree TEM would be a great addition to provide the resolution to resolve the membrane trafficking we observe. However, we do not currently possess that expertise and would need to initiate a new collaboration to do this work.  Again, we see this as a new separate study.

  1. Please provide comment on the specificity of all drugs. Maybe I missed it, my apologies if so.

R: The pharmacological reagents are outlined in Methods under Reagents and Buffers.

  1. Please provide a couple examples of secondary alone staining for confocal. Could be supplementary, just nice to have.

R: This has been done as a standard procedure to test our secondary antibody and the results of control experiments are not included in the manuscript to conserve space. The use of appropriate controls to validate our antibodies has been further emphasized in the methods section. 

  1. Please provide some overlay examples of membrane staining and AQP5 labeling for better visualization of membrane to cytoplasm trafficking

R: For Figures 3, 4, and 6 showing the influx pathway we had considered displaying AQP5 and WGA double labelling, but the inclusion of these extra images reduced the space to display the magnified single labelled images and offered no clear advantage over the use of the single labelled images of AQP5 labelling. However, Figure 7 showing images from the equator efflux zone we have include split image that include single AQP5 areas and adjacent areas labelled with both AQP5 and WGA.

  1. Please provide some rough quantification of all confocals to help assure the reader that the changes observed are meaningful. This could be overall intensity redistribution changes.  It is incredibly difficult to decipher these confocal images for the reader from just looking at the images, even at first glance they are kind of confusing to be honest and I am well-trained in confocal across many cell and tissue types including lenses. Adding some additional labels and outlining structures with annotation will greatly help the interpretation of these figures to the readers. 

R: As outlined above we are somewhat “lucky” that the change to the subcellular location of AQP5 we observe in fiber cell membranes in response to pilocarpine addition occurs in all or nothing fashion only in discrete areas of the lens that means quantification of the shift in membrane localisation not strictly necessary. Despite this we have tried to quantify the confocal images but encountered some technical issues which were difficult to overcome. The membrane label WGA that we used does not label the large gap junction plaques on the broad sides of fiber cells and hence did not produce the complete membrane mask required to define “membrane” and “cytoplasmic” domains for automated analysis of changes in AQP5 labelling between the two pools.

  1. The authors need to expand on those comments of the limitations of the study and discuss how future studies can circumvent these limitations:

-The mechanism of how exactly AQP5 is moved from the membrane is still not known, just that TRPV1 is involved and probably plays a key role as an initiator of some sort.

-It still remains enigmatic if TRPV1 and AQP5 directly interact, or this is an indirect phenomena. Could there be additional mediators required? Please expand on this.

R: While we understand the authors request to discuss the signalling pathways the link TRPV1 activation to changes in the membrane trafficking of AQP5 we feel that such discussion would be premature and therefore too speculative. In our present paper we have present a new phenomenon that have a large impact on overall lens function and don’t want to dilute the discussion by taking a reductionist approach that focusses on the signalling pathways down stream of TRPV1. Such discussion we feel is best left for a later study that specifically attempts to dissect that signalling pathway.

-The chemical modulators could have off-target and probably lens penetrance effects, so a genetic model could be very useful. Please expand on this and provide any drug penetrance data available.

R: Unfortunately, we do not have a KO model to use as a control. We have however in our previous publications showed that all modulators produce similar effects on lens surface pressure in the mouse (Chen et al 2019), bovine (Chen et al 2022) and rat (current study) lens. In addition, we have shown that these same reagents affect the membrane distribution of TRPV1/4 channels in the mouse lens (Nakazawa et al 2021) in lenses either in situ or after removal from the eye. Hence, we are confident there are not penetration problems for the TRPV1 modulators and not off target effects. Since the effects of these reagents on the lens has been published multiple times in the past we do not think penetrance is a problem that warrants further discussion.  

-The effect the findings have on lens transparency or homeostasis is not determined. 

R: We agreed that the effects of the observed redistribution of AQP5 on lens homeostasis have not determined in this current study but we feel that we already adequately discuss what these effects maybe, and how they can be measured in future studies.

-Correlation of findings with non-rodent lenses. How do the findings relate to lens evolution. How do the findings relate to human lens biology?

R: In our previous publication Petrova et al 2020, we pointed out that the rat lens, unlike the human lens, does not accommodate, and discussed that in non-accommodating lenses the observed effects of changing zonular tension on AQP5 membrane trafficking and lens pressure (water transport) serve to provide external regulation of the steady state optics of the lens to ensure light will remain correctly focussed on the retina. Whether in the human lens the dynamic alterations in zonular tension used to initiate lens accommodation also involves changes in AQP5 distribution and water transport remains to be determined. In this regard it is interesting to note that in the human lens, the radius of the anterior lens surface is ~4.7 times greater than the posterior surface, and that the anterior surface changes more than the posterior during accommodation in young human lenses (Dubbelman & Van der Heijde, Vision Res 41:1867-77, 2001). Whether AQP5 plays a role in the process of accommodation in the human lens will be an interesting but difficult area to pursue in future work.

-How do the data relate to patients receiving pilocarpine for presbyopia?

R: Good question. The use of pilocarpine to treat presbyopia relies on the premise that constriction of the pupil changes the depth of focus and allows eyes to focus on near objects in patients who can no longer accommodate. In this view of mechanism of action of pilocarpine no effects on the lens are thought to occur. However, our data on non-accommodating rat lenses would suggest that pilocarpine may also have an additional effect on the steady state optics of the lens through changes to the anterior surface curvature. To address this question, we are currently conducting an in vivo MRI study on presbyopes to look at the effect of pilocarpine application on lens water content and optical power.

-The experiments were performed on young rats, what about aged rats? Is this an age-specific phenomena in rodents?

R: This is an interesting question which can be pursued in future studies. In the current paper we have chosen one age to study a new phenomenon that had not previously been considered. Having now shown that AQP5 trafficking is regulated by zonular tension via a TRPV1-mediated pathway we can now see how this mechanism is altered in not only older animals but also younger ones.  

-How do the data go beyond the lens to understand TRPV1 and AQP biology? This could be interesting to add since pilocarpine is used as a treatment for dry mouth.

-What happens to the overall biology of the lenses beyond just AQP5 with the pharmacological TRP modulation? This could have been addressed by a high-throughput method… (Some theoretically related work: https://www.ncbi.nlm.nih.gov/pmc/articles/PMC8476594/ ). Based on this ref, it is likely that much more than AQP5 is altered!!!! --- This should be commented on, maybe include this reference if you want to, and explain on it. I think it’s entirely ok to say this is a limitation, reiterating that the AQP5 redistribution is a key effect since it happens so quickly., and that future work will capitalize on high-throughput methods to find the full range of TRP channel lens requirements.

-What if the lens was cultured posteriorly with artificial vitreous and anteriorly with artificial aqueous, would that change the findings? (not sure this is possible with today’s tech but interesting to speculate on..)

R: Thank you for these valuable suggestion for future directions for our research, however, we think these future directions are too premature to include in the discussion of our current results.

  1. Please provide a diagrammatic cartoon of the microelectrode equipment and experimental setup to figure 2 to visually explain how the surface pressures was determined. This will be very helpful for other biologists in the TRP field. 

R: Diagrams on the setup (Gao et al 2011) and more in-depth details on the experimental protocols (Figure 1 in Chen et al 2020) have been published previously and we feel do not add significantly to the current manuscript. We have, however, emphasized in the methods the references to these previous studies in the methods section.

  1. The authors should have validated AQP5 redistribution modulation with the various treatments via another protein method, such as western blot or co-ip mass spec from plasma membrane and cytoplasmic fractions of treated lenses. It would be incredibly nice to include this. To be honest, I would be wary of any singular method that shows a redistribution, it is best to orthogonally confirm with another type of related experiment.

R: Again, we agree with the reviewer and are actively working on additional methods to confirm that the observed localised redistribution of AQP5 labelling that we observe in response to changes in zonular tension induced by pilocarpine are indeed caused by changes to the membrane trafficking of AQP5. However, the localised nature of the AQP5 redistribution to specific regions of the lens makes it difficult to isolate cytoplasmic and membrane fractions from the specific regions of the lens that can then be analysed by western blot or co-ip mass. Hence in this manuscript we have focussed more on elucidating the signalling pathway that transduces the changes in zonular tension to the observed redistribution of the AQP5 labelling pattern rather than the membrane trafficking per se.

  1. It would be nice to include a raw data policy, such as available upon request. Up to you but this will become standard sooner than later.

R: We have the raw data which was not required upon submission but can be supplied.

Round 2

Reviewer 2 Report

The authors have addressed sufficiently all inquiries raised by this reviewer during review.